# A Portable Pull-Out Soil Profile Moisture Sensor Based on High-Frequency Capacitance

**DOI:** 10.3390/s23083806

**Published:** 2023-04-07

**Authors:** Zhentao Sheng, Yaoyao Liao, Shuo Zhang, Jun Ni, Yan Zhu, Weixing Cao, Xiaoping Jiang

**Affiliations:** 1College of Agriculture, Nanjing Agricultural University, Nanjing 210095, China; 2National Information Agricultural Engineering Technology Center, Nanjing 210095, China; 3Engineering Research Center of Smart Agriculture, Ministry of Education, Nanjing 210095, China; 4Jiangsu Collaborative Innovation Center for the Technology and Application of Internet of Things, Nanjing 210095, China

**Keywords:** pull-out, soil profile, moisture sensor, high-frequency capacitance, measurement accuracy

## Abstract

Soil profile moisture is a crucial parameter of agricultural irrigation. To meet the demand of soil profile moisture, simple fast-sensing, and low-cost in situ detection, a portable pull-out soil profile moisture sensor was designed based on the principle of high-frequency capacitance. The sensor consists of a moisture-sensing probe and a data processing unit. The probe converts soil moisture into a frequency signal using an electromagnetic field. The data processing unit was designed for signal detection and transmitting moisture content data to a smartphone app. The data processing unit and the probe are connected by a tie rod with adjustable length, which can be moved up and down to measure the moisture content of different soil layers. According to indoor tests, the maximum detection height for the sensor was 130 mm, the maximum detection radius was 96 mm, and the degree of fitting (R^2^) of the constructed moisture measurement model was 0.972. In the verification tests, the root mean square error (RMSE) of the measured value of the sensor was 0.02 m^3^/m^3^, the mean bias error (MBE) was ±0.009 m^3^/m^3^, and the maximum error was ±0.039 m^3^/m^3^. According to the results, the sensor, which features a wide detection range and good accuracy, is well suited for the portable measurement of soil profile moisture.

## 1. Introduction

The root systems of crops are distributed at different depths of the soil, and the soil water content differs at different depths. Therefore, the vertical distribution characteristics of soil water content seriously affect water absorption, growth, and development, and even the final yield of crops [1,2]. To realize intelligent farmland irrigation and a high yield and quality of crops, it is essential to accurately measure the vertical soil profile moisture content in real time. Agriculture currently accounts for 70% of total human water use [3,4]; water shortage has become the primary constraint on agricultural production in many areas of China. The reliable and stable perception of farmland moisture information also plays important roles in guiding and guaranteeing field water management and saving water in agriculture [5,6,7]. Therefore, it is urgent to develop a soil moisture perception method with high precision, low power consumption, high reliability, and easy adjustment through in-depth studies of soil moisture, to expand theoretical knowledge and technological developments, such that a practical and durable soil profile moisture sensor can be developed.

Currently, the mainstream methods for measuring soil moisture content include the oven-drying method, dielectric method, neutronic method, tensiometry method, and remote sensing method [8,9,10,11]. The dielectric method is based on the single-valued function theory proposed by Topp, stating that the soil moisture content is the dielectric constant [12,13]. The soil moisture sensor developed using this method has the advantages of a short measurement time, low cost, and high safety, and it is widely recognized. Moisture sensors based on the dielectric method are divided into needle type and profile type. Among them, the needle type makes direct contact with the soil, which makes it convenient to measure the moisture of the surface soil, and is highly accurate in measurement. However, the typical probe length for the needle-type moisture sensor is 30–200 mm [14], and the root lengths of common food crops exceed 200 mm [15,16]. Therefore, when using a needle-type moisture sensor to measure the moisture content of multiple soil layers, it is necessary to dig up a large soil profile for layered measurement, which makes testing difficult. To address the difficulties encountered when using needle-type moisture sensors to measure multilayer soil water content, soil profile moisture sensors based on the noncontact dielectric measurement principle have attracted widespread attention from scholars at home and abroad.

The soil profile moisture sensor is isolated from the soil through a PVC pipe on the outer wall, and the noncontact measurement of the soil profile moisture is realized by using a single mobile probe or fixed multiple probes. Yuan et al. [17] designed an equidistant three-depth soil profile moisture sensor based on the impedance method, but its measurement radius was only 30 mm, so it was easily affected by the soil gap; following the example of EnviroSCAN, Gao et al. [18] developed a soil profile moisture sensor capable of measuring the moisture content of multiple soil layers and conducted indoor tests. However, since there is no field accuracy verification test, the comprehensive performance indicators must be further demonstrated. Gao et al. [19] designed a soil profile moisture sensor based on high-frequency capacitance, which can be used to determine the water content of an unknown soil layer, but its large volume made it inconvenient to carry and difficult to promote. Deng et al. [20] designed a low-cost, high-precision soil moisture monitoring system with conductivity compensation. With a lower operating frequency, the sensor would be affected by different soil types. An end-cloud-integrated multilayer soil moisture Internet of Things sensor was designed by Shi et al. [21], which was suitable for the long-term monitoring of soil profile moisture, but this was expensive because of its complex circuitry. Escriba et al. [22] developed a new generation of affordable capacitive sensors compatible with Toward v4.0 Agriculture; however, the sensors were connected to concentrators using cables and were not convenient for deployment in the field. Lloret et al. [23] designed a soil moisture sensor based on a coil and discussed the feasibility of using the sensor to measure the soil profile, but the measurement accuracy of the sensor needs to be further verified through performance testing. In addition, foreign well-known commercial soil profile moisture sensors, such as TRIME-PICO-T3 [24] in Germany, PR2 [25] by Dalta-T in the UK, and EnviroSCAN [26] by Sentek in Australia, cannot be widely promoted and applied in the agricultural field due to their high prices [27,28].

In this study, a low-cost, single-probe pull-out soil profile moisture sensor was designed following the principle of the capacitive dielectric method, for which the length of the tie rod can be flexibly configured to move it up and down a single-ring probe to measure the moisture of different soil layers, and a smartphone app is used to read the moisture content and other moisture information for the soil in real time. Compared with current moisture sensors, such as those based on time domain reflectometry (TDR) and frequency domain reflectometry (FDR), it does not require high-precision measurement circuitry and a complex secondary calibration process [29,30,31,32], making it very suitable for portable mobile measurements of soil profile moisture. Some of the innovations of the designed soil profile moisture sensor are as follows: (1) the sensor can achieve the fast in situ detection of soil profile moisture requirements because of its ability to sense moisture by using an electromagnetic field; (2) the sensor is portable and low-cost, consisting of only one movable probe that detects the moisture of different soil layers, and the data can be transmitted to a smartphone for display without additional hardware.

## 2. Design of the Soil Profile Moisture Sensor

### 2.1. The Measuring Principle

Soil is a mixture of dielectrics, and the dielectric constant of water is higher than the dielectric constant of other solid particles by a quantity product (as shown in Table 1). The soil moisture content I can be derived by measuring the dielectric constant of the soil (as shown in Equation (1)). The main methods for measuring soil moisture content using the dielectric method are the TDR, FDR, and capacitance methods. However, the signal generation and processing circuits of TDR and FDR moisture sensors are extremely complex, resulting in high prices for the equipment produced, making them unsuitable for widespread application. The capacitive method mainly uses a resonant circuit to generate an oscillation signal to excite the measurement probe, and the moisture sensor based on this principle is easy to make, is simple to use, offers a high performance–cost ratio, and is more suitable for the majority of farmers.

For the probe based on the capacitance measurement principle, the probe and the soil form an equivalent capacitance C (as shown in Figure 1), and there is a functional relationship between the equivalent capacitance and the soil permittivity ε (as shown in Equations (2) and (3)). Thus, the soil moisture content can be indirectly calculated by measuring the size of the equivalent capacitance value.
(1)$$\theta_{v}=-5.3\times 10^{-2}+2.92\times 10^{-2}\varepsilon_{r}-5.5\times 10^{-4}{\varepsilon_{r}}^{2}+4.3\times 10^{-6}{\varepsilon_{r}}^{3}$$
(2)$$C=\frac{\varepsilon S}{4\pi kd}$$
(3)$${\varepsilon =\varepsilon }_{0}\varepsilon_{r}$$

Among them:

$\theta_{v}$—Soil volume moisture content (m^3^/m^3^);

ε—Soil permittivity (F/m);

$\varepsilon_{r}$—Relative permittivity (dimensionless);

$\varepsilon_{0}$—Vacuum permittivity ($\varepsilon_{0}=8.85\times 10^{-12}$ F/m);

*C*—Equivalent capacitance consists of probe and soil (F);

*S*—Area of parallel plates (m^2^);

*d*—Space between two parallel plates (m).

The existing capacitive moisture sensor is mainly based on the RC peak detection principle: The soil moisture content changes the equivalent capacitance C3 value by affecting the dielectric constant of the high humidity-sensitive polymer membrane of the probe. The equivalent capacitance is charged and discharged based on the multivibrator of the 555 chip [33], and the peak voltage V_out_ of the output is measured to calculate the soil moisture content (as shown in Figure 2). However, the low resonant frequency of this type of moisture sensor leads to high dielectric loss, which can cause the circuit to operate unstably (as shown in Figure 3a).

In this study, an LC oscillation circuit based on the MC12149 chip [34] was used to generate a resonant signal of about 200 MHz to excite the ring probe (shown in Figure 4). Under the action of the excitation source, the probe radiates the electromagnetic field to the soil to form an equivalent capacitance C5. The equivalent capacitance and the LC resonant circuit produce a resonant frequency *f* that varies as the soil moisture content changes (shown in Equations (4)–(6)). By fitting the relationship between this frequency and soil moisture, the soil moisture content under any condition can be inverted. The LC high-frequency resonant capacitor method can not only reduce the dielectric loss (shown in Equation (7)) but also ensure consistency between the soil moisture measurement results of different soil types because the polarization intensity of the soil medium is a function of frequency (as shown in Figure 3b).
(4)$$f=\frac{1}{2\pi \sqrt{LC}}$$
(5)$$L=L_{p}+L_{2}$$
(6)$$C=\frac{{(C}_{5}+C_{p})\times C_{4}}{{(C}_{5}+C_{p})+C_{4}}$$

Among them:

f—The resonance frequency of the soil equivalent capacitance (Hz);

L—Total resonant inductance (H);

C—Total resonant capacitance (F);

$L_{p}$—Circuit parasitic inductance (H);

$L_{2}$—Body resonant inductance (H);

$C_{5}$—Soil equivalent capacitance (F);

$C_{4}$—Decoupling bias capacitors (F).
(7)$$\varepsilon \left(\omega \right)=\varepsilon^{\prime }\left(\omega \right)-j\varepsilon^{\prime \prime }\left(\omega \right)$$

Among them:

$\varepsilon \left(\omega \right)$—Plural form of dielectric constant;

$\varepsilon^{\prime \prime }\left(\omega \right)$—Dielectric loss;

$\varepsilon^{\prime }\left(\omega \right)$—The degree of polarization of the medium.

### 2.2. Hardware Circuit Design

The hardware circuit of the pull-out soil profile moisture sensor includes two parts: a moisture-sensing module and a data processing module. The overall block diagram of the hardware circuit is shown in Figure 5. The moisture-sensing module converts the soil moisture content into a frequency signal, consisting of a ring probe, a voltage-controlled oscillation circuit, and a frequency dividing/conditioning circuit. The data processing module measures the frequency signal and transmits the converted moisture content value to the smartphone app using the Wi-Fi communication protocol, which consists of the control circuit, wireless communication circuit, and power supply circuit.

#### 2.2.1. Moisture-Sensing Module Design

The ring probe selects a double copper ring with a diameter of 40 mm, a height of 30 mm, and a spacing of 20 mm as the upper and lower plates of the soil-equivalent capacitor C5. The two plates are connected to the MC12149 voltage-controlled oscillator using the solder joint to form a capacitor three-point LC oscillation circuit, which has a short oscillation time and high oscillation resonance frequency. Since the MC12149 chip has its own buffer amplifier, it can be connected to the poststage divider circuit without adding a power amplifier circuit. The reason for designing the frequency dividing circuit is that the single-chip microcomputer cannot directly measure the high-frequency signal and the frequency dividing circuit includes a 64-way circuit based on the MB504 chip and a 256-division circuit based on the 74H393 chip. In order to avoid the influence of the output signal of the frequency dividing circuit by the poststage digital circuit. The signal conditioning circuit is designed that is a photoelectric coupling circuit, which is used to isolate the output of the frequency dividing circuit and the input of the single-chip microcomputer. The hardware circuit of the moisture-sensing module is shown in Figure 6.

#### 2.2.2. Data Processing Module Circuit Design 

The processor of the data processing module is the ATmega328P microcontroller [35], which uses an external 16 MHz crystal as a high-precision clock source. The embedded Arduino ISP firmware completes the configuration of the base clock. The sensor was developed using external interrupts, UARTs, and ADC peripherals of the Atmega328P microcontroller. Among them, the external interrupt is used to respond to the trigger of the moisture probe output signal and assist in the measurement of the frequency signal and the conversion of soil moisture content, the ADC peripheral realizes the detection of the power of the 26650-lithium battery, and the UART peripheral uses the AT command to complete the communication with the Wi-Fi module. The Wi-Fi module selects the ESP8266 model, the receiving antenna is a PCB on-board antenna, and a pair of NPN transistors are used to achieve level conversion with the single-chip UART. The power supply circuit is boosted and stepped down to power the circuit of each module of the system, with a 5 V output to power the single-chip microcomputer and probe and a 3.3 V output to power the Wi-Fi module. The hardware circuit of the data processing module is shown in Figure 7.

### 2.3. Software Design

The software program consists of the upper computer data receipt and display program as well as the lower computer measurement program. The lower computer is developed based on the Arduino platform. The software includes four parts: soil moisture measurement program, battery power detection program, system power management program, and data package sending program. The upper computer is developed based on the Eclipse platform and Java language, and the software includes a communication connection program, data decoding program, and interface display program. The system block diagram is shown in Figure 8.

The soil moisture measurement program is the core of the embedded program of the lower computer, which measures the frequency of the output signal of the moisture-sensing probe and converts the measured frequency value into the actual soil moisture content value by using the constructed moisture measurement model. The specific process of measuring the frequency signal of the single-chip microcomputer is shown in Figure 9. First, the rising edge of the frequency signal triggers the external interrupt of the single-chip microcomputer. Second, the internal timer is cleared, and the TCNT1 counter starts counting. Third, when the next pulse rises on edge, the function is used in the program to read and store the TCNT1 value. Finally, the frequency value is calculated according to Equation (8). The fuel detection program collects the lithium battery voltage and, when set outside the on-chip ADC, calculates the voltage based on the number of ADC samples. The power management program drives the sensor into hibernation after a period of operation (when the battery is low) [23] to increase the total operating time of the sensor when fully charged and prevent the battery from being damaged due to overdischarge. The primary function of the data packaging transmitter is to integrate the frequency value and power value measured by the moisture sensor into a string and drive the ESP8266 communication module to transmit it in the form of electromagnetic waves using the AT command of the single-chip microcomputer.
(8)$$f=\frac{f_{Y}}{T_{c}}$$

Among them:

f—The output frequency of moisture-sensing module (Hz);

$f_{Y}$—The main frequency of moisture-sensing module (Hz);

$T_{c}$—The value of TCNT1 counter microcontroller (dimensionless).

Figure 10 is the display interface and timing diagram of the upper computer app. The main body of this app is developed in Java language and, at the same time, uses Edit Text, Text View, Button and Progress Bar, Text View, and four Eclipse graphical plug-ins for auxiliary programming, of which the Text View plug-in is used to display the soil moisture content value. The Edit Text plug-in is used to enter sensor ID information, and the Button plug-in is used to confirm the network connection. The Progress Bar plug-in is used to display battery-level information. After the smartphone app establishes a connection with the moisture sensor, it receives and parses the data package uploaded by the sensor in real time. The process of the frequency value and voltage value into moisture content value and power value by the moisture measurement model obtained through testing and the lithium battery power discharge model provided by the manufacturer presents the transformed data as a string in the interface. The app’s detailed procedure is presented as follows: First, use the mobile phone to connect to the Wi-Fi hotspot of the sensor. Second, enter the ID information of the sensor in the app, and click the OK button. Finally, in real time, you can view its electricity and soil moisture content measured by the moisture sensor.

### 2.4. Shell and Pull-Out Structure Design

Due to the harsh operating environment of soil profile moisture sensors, the circuitry of the sensor must be protected to improve its tightness and insulation level under the condition that the performance is only limited to influence and interference. This study specially designs and develops a set of sensor housing protection and segmental pull-out positioning devices based on Solidworks 3D modeling software (shown in Figure 11). The device consisted of two parts: a data processing unit and a moisture-sensing probe, including central parts, such as the main board cavity, card slot, probe bracket, probe tube sleeve, segmented tie rod, and isolation tube. The sensitive copper ring element is mounted on the probe bracket, and the two brackets are fixed by the PCB circuit board and sealed by the probe sleeve. The data processing unit and the moisture-sensing probe are physically connected using tie rods, and the FPC line is used to realize the electrical connection of the two ends of the circuit in the hollow tie rod.

This set of devices has three advantages: (1)The sensor tie rod is designed as a segmental type, which is convenient to freely increase or decrease the number of segments according to the soil depth measured according to needs.(2)The positioning card ring is designed with a specially shaped card slot that matches the segment (this sensor is diamond-shaped), and the automatic positioning of the probe can be realized by simply raising the height of the tie rod to the corresponding soil depth when measuring, and then rotating a certain angle for misalignment.(3)They draw on mineral water bottles and design the protective shell of the data processing unit into a large and small middle at both ends and a profile structure that is easy to grasp.

## 3. Sensor Performance Tests

### 3.1. Materials and Methods

#### 3.1.1. Sensor Usage

Figure 12a is a physical product image of the developed pull-out soil profile moisture sensor. The detailed procedure is shown in Figure 12b. First, the soil extractor is used to bury the PVC isolation pipe in the soil, and the sensor is put into the PVC pipe to achieve noncontact measurement. Second, the moisture-sensing probe converts the soil moisture into a frequency signal using an electromagnetic field and transmits it to the data processing unit. Third, the data processing unit completes the measurement of the frequency signal, calculates the moisture content value according to the constructed soil moisture measurement model, and then sends the moisture content value out through the Wi-Fi protocol. Finally, the user connects the sensor to a Wi-Fi hotspot to view the corresponding soil moisture value on the smartphone app. By twitching the segmented tie rod above the card slot, the probe hovers to sense the moisture content of different soil layers, and the length of the tie rod can be flexibly adjusted.

The sensor is more suitable for portable measurements in multiple plots (as shown in Figure 12c, i.e., PVC pipe is already embedded in each field community). The single sensor is placed into each PVC pipe in turn to complete the measurement of water content of multiple soil layers in different plots. This is a compromise method that can not only measure quickly, but also minimize the cost of the experiment to the greatest extent.

#### 3.1.2. Experimental Design

A performance test of the portable pull-out soil profile moisture sensor was carried out in Nanjing, Jiangsu Province (31°65′ E, 119°02′ N) (Figure 13), and the performance test of the sensor consisted of two parts: a laboratory test and a field test. The laboratory test mainly includes three parts: maximum detection range test, calibration of soil volumetric moisture content, and measurement accuracy verification test. The field test aims to verify the measurement accuracy of the sensor in the field. The laboratory test procedure is as follows: loam soil samples were collected in Jiangsu Province; then the soil samples taken from the field were removed from weeds, air-dried, ground, screened (2 mm), dehydrated, and put into a large barrel with a lid for storage.

The sensor is placed in the inner circle PVC pipe during the test, and the soil sample is filled in the outer circle PVC pipe. The test barrel (shown in Figure 14) is specially designed to improve the accuracy of the test. A PVC pipe with a diameter of ∅50 mm and a height of 500 mm is fixed in the center of the lower round bottom cover as the inner pipe. Then, a larger PVC pipe is sheathed on the lower bottom cover as the outer wall to form another annular structure.

(1)Maximum Detection Range Test

The following tests were designed to obtain the sensor probe’s maximum detection diameter and maximum detection height. Among them, the test scheme of the maximum detection diameter was as follows: first, add dehydrated soil samples to 7 test barrels with different outer diameters (i.e., ∅75, ∅90, ∅110, ∅125, ∅140, ∅160, ∅200 mm, shown in Figure 15; this PVC pipe with standard outer diameter is easy to obtain), and then insert the sensor into each test barrel filled with soil to measure and record the corresponding probe output frequency in turn, and draw the curve of the sensor probe output frequency with the diameter of the soil sample after the test is completed. Next, take 0.5 times the abscissa with a slope of 0 as the maximum detection radius of the sensor probe. The test scheme of the maximum detection height is as follows: Place 16 dehydrated soil samples with different height differences spaced 10 mm apart sequentially in a sufficiently large test barrel (soil sample interval is (0,16)), and then use the sensor to measure and record the probe output frequency of soil samples at different heights. After the test is completed, the output frequency of the sensor probe is drawn with the height of the soil sample, and the abscissa value when the slope is 0 is taken as the maximum detection height of the sensor probe.

(2)Soil Volumetric Moisture Content Calibration

As shown in Figure 16, the ground soil particles are added to each mixing drum, and a certain amount of water is poured into each measuring cup. After stirring well, it is configured into nine soil samples with obvious gradient moisture content and loaded into a sufficiently large test barrel (spread every 50 mm and properly compacted), sealed with plastic wrap at room temperature for 48 h. First, the self-developed sensor was used to measure the soil samples with different moisture content, and the corresponding output frequency of the sensor was recorded separately. Then the soil was sampled using a fixed-volume ring knife, the sampled soil was dried in the oven, the quality of the soil sample before and after drying was recorded using the balance, and the volume moisture content of the corresponding soil sample was calculated and recorded according to the oven-drying method. After the test is completed, the relationship curve between the output frequency of each soil sample of the sensor and the actual volumetric moisture content is plotted, and the mathematical equation represented by the curve is the moisture measurement model of the studied sensor.

(3)Laboratory Verification Test of Measurement Accuracy

Manually configure 35 soil samples with unknown moisture content with an obvious gradient; the moisture content interval was [0, 0.50 m^3^/m^3^]. Put the soil samples into a large-enough transformation test barrel, and then store in a sealed barrel for 48 h at room temperature and measure the soil samples using self-developed moisture sensors. The actual volumetric moisture content of soil samples was measured using the oven-drying method and commercial moisture instrument WET-2. Using the value as a reference, and the root mean square error, the mean deviation error and maximum error of the laboratory measurement results of the self-developed sensor were calculated to verify the laboratory measurement accuracy of the sensor.

(4)Field Verification Test of Measurement Accuracy

Two plots with obvious moisture gradients were selected outdoors, four test points were selected each, a hole with a diameter of 50 mm and a depth of about 400 mm was dug at each test point using a soil extractor, the bottom-sealed PVC pipe was inserted into each hole, and the top of the PVC pipe was sealed with a lid. After standing for 48 h, the self-developed sensor was inserted into the PVC pipe. The moisture content of different soil layers (such as 0–100 mm, 100–200 mm, and 200–300 mm) was measured by pulling the tie rod up and down. A total of 24 soil samples of unknown moisture content were tested. As a control, a soil profile was dug near the test point (as shown in Figure 17), and the moisture content of the corresponding soil layer was measured using a purchased commercial moisture sensor, WET-2 (Delta-T Company in the UK). Using the value as a reference, the root mean square error, mean deviation error, and maximum error of the field measurement results of the self-developed sensor were calculated to verify the field measurement accuracy of the sensor.

#### 3.1.3. Data Processing Methods

(1)Drying method for measuring soil volumetric moisture content

The drying method is the main means of measuring the soil volume moisture content value and mass water content. The detailed procedure of the drying method is presented as follows: First, use a ring knife to take a fixed volume (*V*) soil sample, and put it into an aluminum box; weigh out the total weight (*M*). Second, put the aluminum box with the lid removed and put it into the drying box for drying, take it out, and weigh it again as *m*. Finally, *ρ_w_* represents the density of water in the soil (1 at room temperature), ∆*M* represents the reduced mass of the soil after drying, and the volumetric moisture content of the soil (*Ɵ*) is calculated according to Equation (9).
(9)$$\Theta =\frac{∆M}{V}$$
(10)$$∆M=\frac{M-m}{\rho_{w}}$$

(2)Sensor detection radius

Figure 18 shows the relationship curve of the sensor output frequency with the measured soil sample radius (or height), where the abscissa is the radius or height of the soil sample to be measured, and the ordinate is the sensor output frequency value. Find the abscissa with a slope of zero on the curve (beyond which the output frequency value of the sensor will not change), and 1/2 of this abscissa value is the detection radius of the sensor. The calculation is performed according to Equations (11) and (12), where *L* is the maximum detection radius or height of the sensor, $S_{n}^{\prime }$ is the slope of the curve, and *x* and *y* are the horizontal and vertical coordinates of the response curve, respectively.
(11)$$L={X|}_{S_{n}^{\prime }=0}$$
(12)$$S_{n}^{\prime }=\frac{∆y}{∆x}$$

(3)Sensor measurement accuracy evaluation indexAccording to the relevant literature and product specifications of moisture sensors [36,37], the measurement accuracy evaluation indicators of moisture sensors mainly include root mean squared error, mean bias error, and maximum error.(1)Root mean square error (RMSE)RMSE reflects the difference between the measured value of the moisture sensor and the actual moisture content value of the soil. RMSE is calculated according to Equation (13), where *N* is the sample size, *M* is the measured (soil sampling) value, *P* is the predicted (sensor measurement) value, and *M* is the average measured value.
(13)$$RMSE=\sqrt{\frac{1}{N}\sum_{i=1}^{N}{\left(M_{i}-P_{i}\right)}^{2}}$$(2)Mean bias error (MBE)The mean bias error (MBE) is the sum of the deviations of the individual measurements of the moisture sensor from the actual moisture content value, divided by the number of measurements, representing the difference of any value in the moisture content measurement of the soil sample. MBE is calculated according to Equation (14), where *N* is the sample size, *M* is the measured (soil sampling) value, *P* is the predicted value (sensor measurement), and *M* is the average measured value.
(14)$$MBE=\frac{1}{N}\sum_{i=1}^{N}{\left(P_{i}-M_{i}\right)}^{2}$$(3)Maximum errorThe maximum error of the sensor refers to the maximum fluctuation between the measured value of the sensor and the actual value, which is used to reflect the worst-case scenario of the measurement, and the calculation is carried out according to Equation (15), where *M* is the measured (soil sampling) value and *P* is the predicted (sensor measurement) value.
(15)$$\xi =max\left|M_{i}-P_{i}\right|$$

### 3.2. Test Results and Analysis

#### 3.2.1. Maximum Detection Range

According to the test method of the maximum detection range in Section 3.1.2, using the least squares method to analyze the obtained experimental data, the relationship curve between the output frequency value of the studied sensor and the height (and diameter) of the measured soil sample was obtained as shown in Figure 19. It can be seen that with the increase in height and diameter, the output frequency value shows a monotonic decreasing trend. Among them, the curve equation of output frequency with soil sample diameter is $y=5\times 10^{-5}x^{2}-0.0198x+15.133$, R^2^ is 0.9928; the curve equation of output frequency with soil sample height is $y=9\times 10^{-5}x^{2}-0.0253x+14.953$, R^2^ is 0.9949; according to Equations (11) and (12), the maximum detection radius of the sensor is 96 mm, and the maximum detection height is 130 mm.

#### 3.2.2. Moisture Measurement Model

The moisture measurement model of the self-developed sensor shown in Figure 20 was obtained using the least squares method to analyze moisture measurement values of the nine soil samples in Section 3.1.2 with the corresponding probe output frequency values. As a result, it can be seen that with the increase in volumetric moisture content, the output frequency of the moisture sensor gradually decreases and the model’s equation is $y=0.0758x^{2}-2.1834x+15.6751$, and the R^2^ is 0.9719.

#### 3.2.3. Laboratory Measurement Accuracy

According to Equations (13)–(15), the root mean square error (RMSE), mean bias error (MBE), and maximum error of the laboratory measurement results of the self-developed moisture sensor are calculated with reference to the oven-drying method. The measurement value of the commercial instrument WET-2 and the results are shown in Table 2. The measured values of the oven-drying method are used as a reference; it can be seen from the table that the root mean square error of the indoor working measurement of the sensor is 0.017 m^3^/m^3^, the average deviation error is ±0.033 m^3^/m^3^, and the maximum error is ±0.031 m^3^/m^3^. Taking the WET-2 measurement as a reference, the root mean square error of the indoor operating measurement of the sensor is 0.018 m^3^/m^3^, the average deviation error is ±0.050 m^3^/m^3^, and the maximum error is ±0.033 m^3^/m^3^.

As shown in Figure 21, taking the sensor measurement as the predicted value, the drying method, and the WET-2 measurement as the actual value, the R^2^ of the curve obtained by linear fitting of the two is 0.9839 and 0.9802, respectively. The comparison results between the self-developed moisture sensor and the drying method and WET-2 are minimal. The WET-2 measurement and the drying method measurement are linearly fitting, and the R^2^ of the curve is 0.9949, which is extremely correlated. The absolute error distribution between the measured value and the actual value of the sensor is characterized by being large at both ends and small in the middle.

#### 3.2.4. Field Measurement Accuracy

Taking the measured value of WET-2 of commercial instruments as a reference, the root mean square error (RMSE), mean bias error (MBE), and maximum error of the indoor measurement results of the self-developed moisture sensor are calculated according to Equations (13)–(15). The results show that the root mean square error (RMSE) of the self-developed sensor measurement results is 0.020 m^3^/m^3^, the mean bias error (MBE) is ±0.009 m^3^/m^3^, and the maximum error is ±0.039 m^3^/m^3^ (shown in Table 3).

As shown in Figure 22, taking the sensor measurement as the predicted value and the WET-2 measurement value as the actual value, the R^2^ between the two is 0.9066. The absolute error distribution between the sensor measurement value and the actual value shows the characteristics of the large right and small left ends.

## 4. Discussion

The development of portable and practical soil profile moisture sensors is a necessary means to improve the efficiency of soil moisture measurement and reduce the cost of soil moisture measurement, especially the application of soil moisture measurement in the multiple plots [38,39,40]; the sensor developed in this study is suitable for the portable and low-cost measurement of soil moisture. The current fixed soil profile moisture sensors on the market are expensive. The prices reach thousands of US dollars, and more sensors need to be deployed when the spatial variation in the soil moisture content in the community is large, making the cost of testing very high. However, if a PVC pipe is embedded in each community in advance, the measurement sensor only needs to be put into the PVC pipe of each community at a specific time. The probe can be pulled up and down to complete the measurement of the depth of water content of different soil profiles in different plots, and both the cost of equipment and the difficulty of testing are significantly reduced.

In the laboratory test, the absolute error distribution between the measured value of the sensor and the actual value shows the characteristics of being large at both ends and small in the middle. The analysis shows that the Jiangsu brown loam used for a laboratory test is a sample with different water content artificially configured after grinding, which is affected by the physical and chemical properties of the loam [41,42]. As a result, the measurement model of self-developed sensors cannot be well adapted to the special situation of high and low moisture content, and subsequent research needs to make appropriate compensation for both ends of the measurement model obtained using data processing. The particle void in the sample soil under low water content conditions is large, and the soil sample volume shrinks rapidly under high moisture content conditions. In addition, the results of field tests are similar to those of laboratory tests, and the relative error also shows the characteristics of large measurement errors in soil with high moisture content.

The linear fitting curve R^2^ of the predicted value and the true value of the sensor in the field environment are lower than those in the indoor environment. The analysis shows that, in the field test environment, a soil extractor needs to be used to dig a hole, the embedded PVC pipe often has a certain air gap with the surrounding soil, and the electromagnetic wave intensity is attenuated exponentially in the air [43]. If the air gap is too large, it is necessary to fill the mud to exhaust the air, and it generally needs to be left to stand for 2 or 3 weeks to ensure that the mud and the soil moisture to be measured are completely penetrated. If the mud structure is different from the soil structure that is to be measured, new errors will be introduced, so the grouting soil should be sampled as close to the test site as possible [44,45].

## 5. Conclusions

In this study, a portable pull-out soil profile moisture sensor was developed based on the high-frequency capacitance method. This work was mainly innovative in terms of the design of the sensor probe structure, the design of the data processing unit software and hardware, the shell and pull-out structure design, and the sensor performance testing and data processing.

(1)The design of the sensor probe adopted an upper and lower ring electrode layout, which, together with the soil, forms a three-point LC resonant circuit equivalent capacitance. The MC12149 resonant circuit oscillates at 200 MHz, which reduces the influence of loamy-type soil on sensor measurements. The resonant frequency is divided twice by the frequency dividing/conditioning circuit, and finally, a low-frequency square wave signal that is easy to measure is obtained.(2)The data processing unit uses the Atmega328P microcomputer to measure the output frequency of the moisture sensor probe and communicates with the display terminal through the ESP8266 module. The lower computer uses the Arduino language for embedded system development based on the Arduino IDE environment, the upper computer uses the Java language for mobile app development based on the Eclipse environment, and the upper and lower computers use the Wi-Fi protocol for communication.(3)Using Solidworks to design segmental tie rods and gourd-shaped housings, the shell of the data processing unit was designed as a gourd-shaped package for easy grip, and the segmental tie rod and the card slot were misaligned to realize the hover measurement of the moisture-sensing probe. The tie rod and the card slot were used in order to enable the moisture-sensing probe to move up and down and allow the user to flexibly configure the segmented tie rod with different lengths according to the depth of the soil to be measured.(4)The data processing results for the laboratory soil sample testing of the sensor show that the maximum detection height of the sensor is 130 mm and the maximum detection radius is 96 mm. The equation of the moisture measurement model calibrated to obtain the sensor is $y=0.0758x^{2}-2.1834x+15.6751$, where R^2^ is 0.9719. The results of the laboratory and field accuracy verification test show that the root mean square error of the sensor was 0.020 m^3^/m^3^, the average deviation was ±0.009 m^3^/m^3^, and the maximum error was ±0.039 m^3^/m^3^.

(5)According to the accuracy evaluation criteria for the soil moisture sensor (RMSE < 0.035 m^3^/m^3^; MBE = ±0.02 m^3^/m^3^), and upon comparing the measurement accuracy of the developed sensor with that of the commercial sensor WET-2 (maximum error = ±0.03 m^3^/m^3^) using a similar measurement principle, it can be seen that the sensor studied in the project has high measurement accuracy and low production cost.

In future work, we would like to perform more practical experiments with more different types of soil to verify the measurement accuracy of the proposed sensor. Then, we would be devoted to promoting the application of the pull-out sensor for the portable soil profile moisture measurement of multiple plots with precision agriculture practitioners.

## Figures and Tables

**Figure 1 sensors-23-03806-f001:**
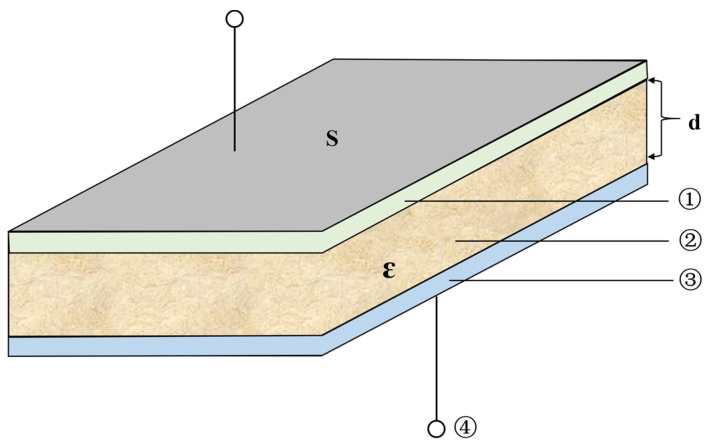
Schematic diagram of the equivalent capacitor structure composed of soil and probe. ① Upper plate; ② soil; ③ lower plate; ④ leads.

**Figure 2 sensors-23-03806-f002:**
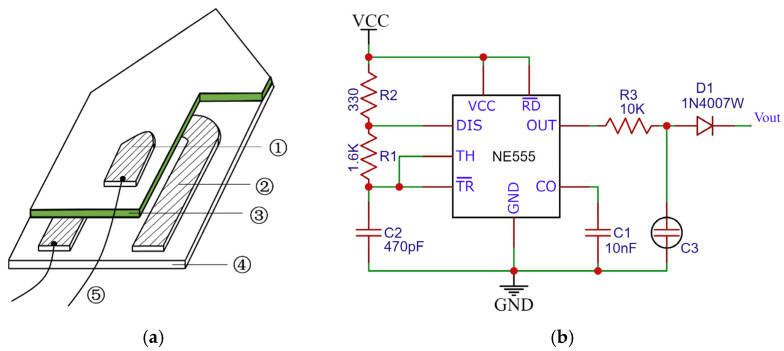
Structure and principle of RC resonant capacitor probe. (**a**) Structural diagram; (**b**) circuit diagram. ① Upper plate; ② lower plate; ③ high humidity-sensitive polymer membrane; ④ substrate; ⑤ leads.

**Figure 3 sensors-23-03806-f003:**
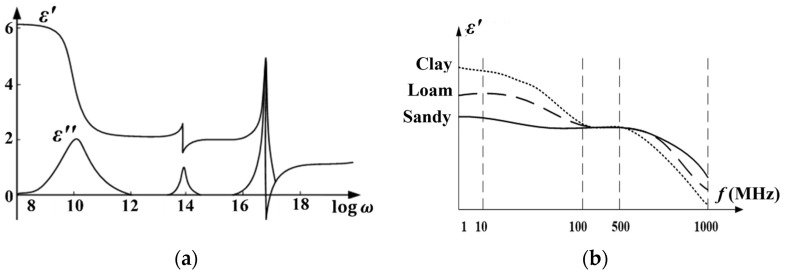
Dielectric constant frequency characteristics. (**a**) Dielectric loss; (**b**) loamy differences.

**Figure 4 sensors-23-03806-f004:**
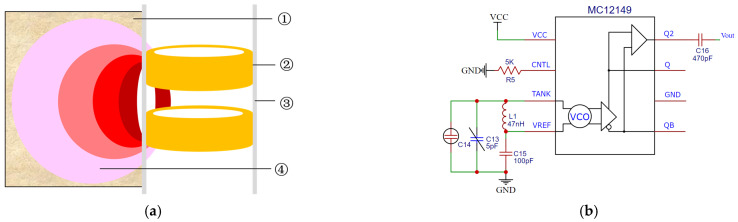
Structure and principle of high-frequency LC resonant capacitor probe. (**a**) Structural diagram; (**b**) circuit diagram. ① Soil; ② copper ring; ③ PVC pipe; ④ electromagnetic field.

**Figure 5 sensors-23-03806-f005:**
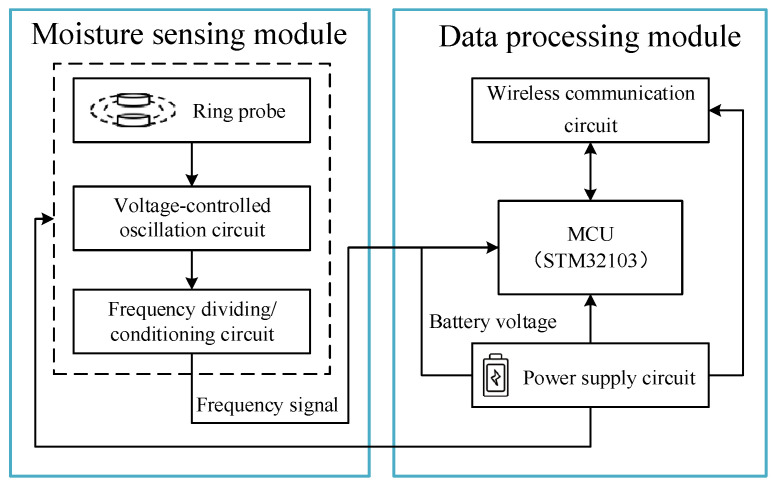
Overall hardware circuit diagram.

**Figure 6 sensors-23-03806-f006:**
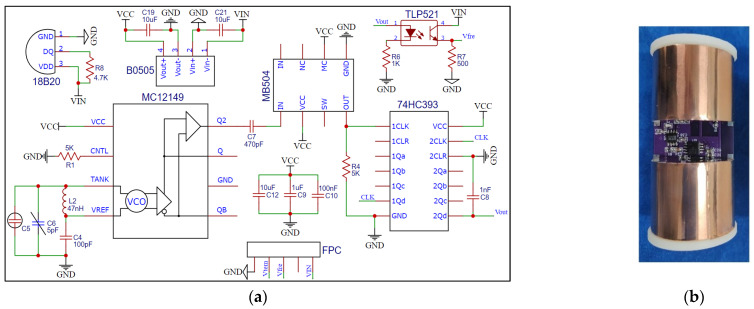
Hardware circuit of the moisture-sensing module. (**a**) Circuit schematic; (**b**) PCB and physical drawings.

**Figure 7 sensors-23-03806-f007:**
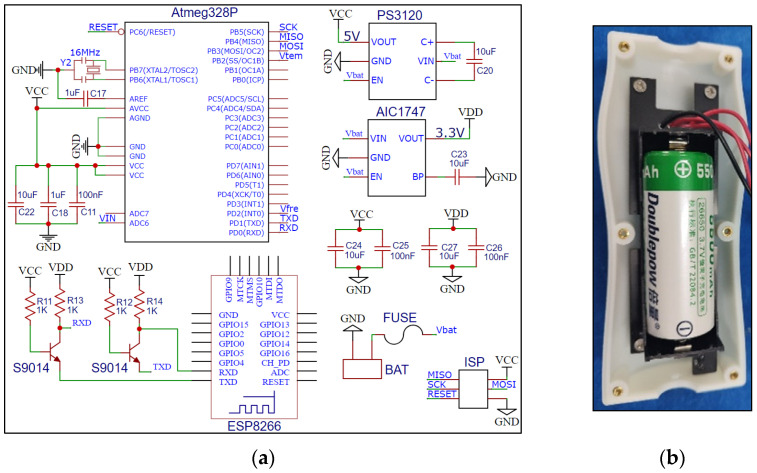
Hardware circuit of data processing module. (**a**) Circuit schematic; (**b**) PCB and physical drawings.

**Figure 8 sensors-23-03806-f008:**
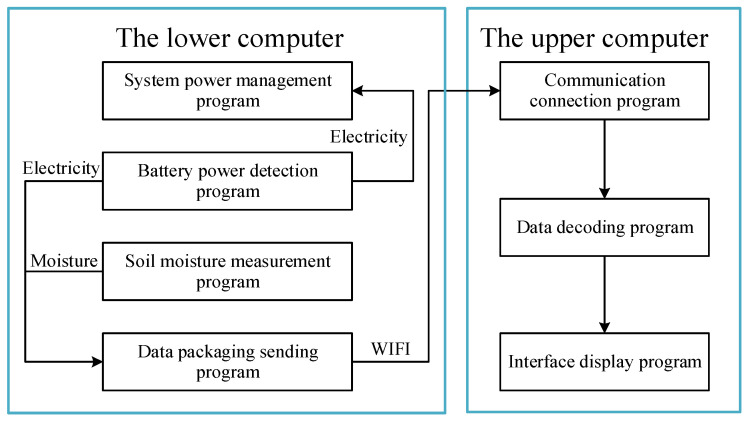
Block diagram of the system software.

**Figure 9 sensors-23-03806-f009:**
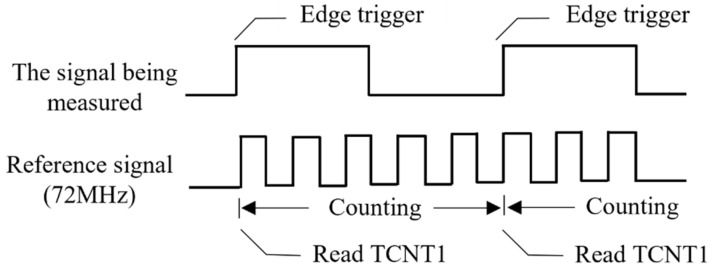
Frequency measurement timing diagram.

**Figure 10 sensors-23-03806-f010:**
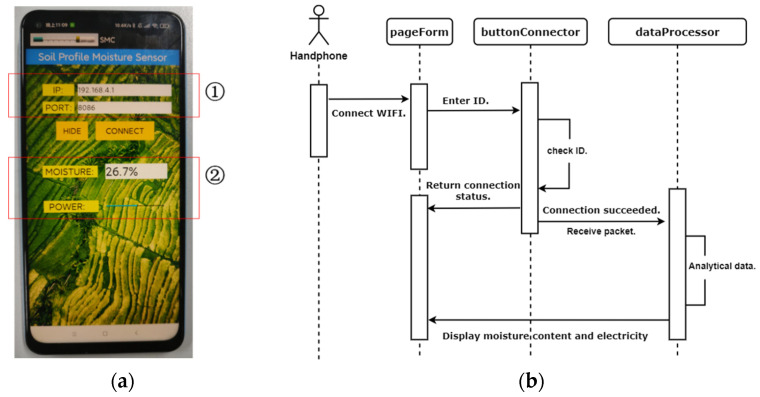
Smartphone app of sensor. (**a**) Interface diagram; (**b**) timing diagram. ① Sensor identification; ② measurement results.

**Figure 11 sensors-23-03806-f011:**
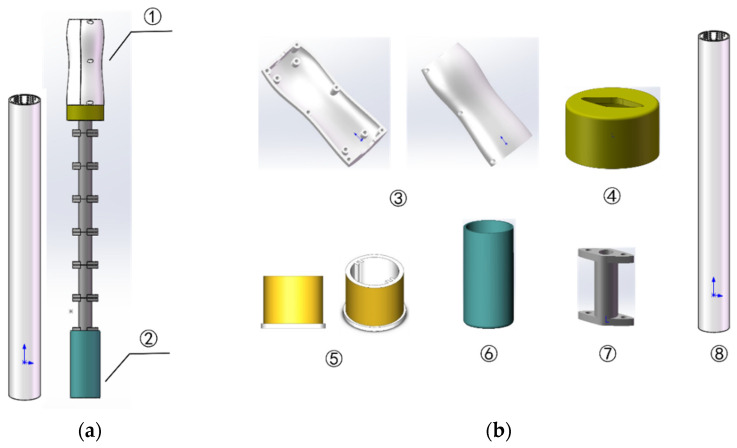
Sensor housing structure. (**a**) Assembly drawings; (**b**) component drawing. ① Isolation tube data processing unit; ② moisture-sensing probe; ③ upper and lower chambers of the motherboard; ④ card slot; ⑤ probe upper and lower brackets; ⑥ probe sleeve; ⑦ segmented tie rod; ⑧ isolation tube.

**Figure 12 sensors-23-03806-f012:**
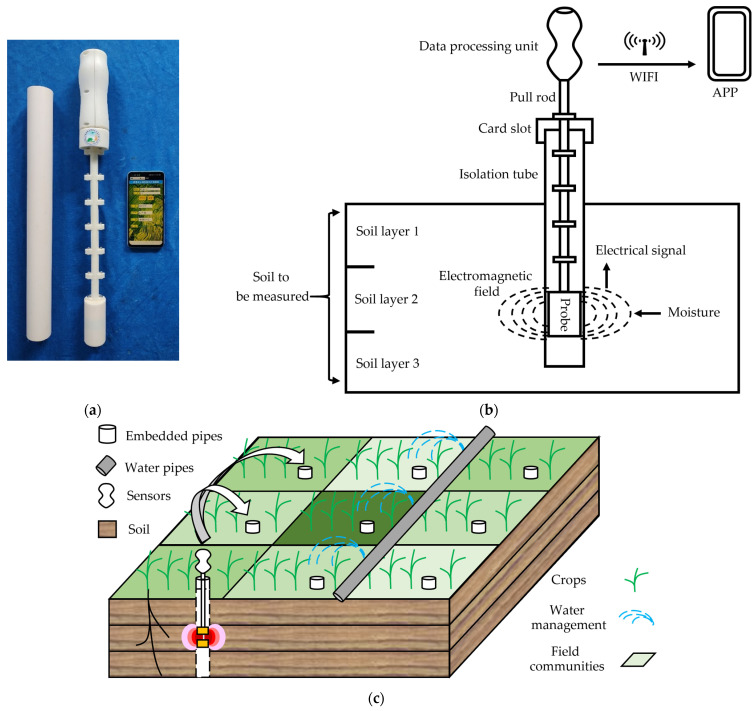
Use of portable pull-out soil profile moisture sensor. (**a**) Physical diagram of the sensor; (**b**) schematic diagram of sensor operation; (**c**) typical application scenarios.

**Figure 13 sensors-23-03806-f013:**
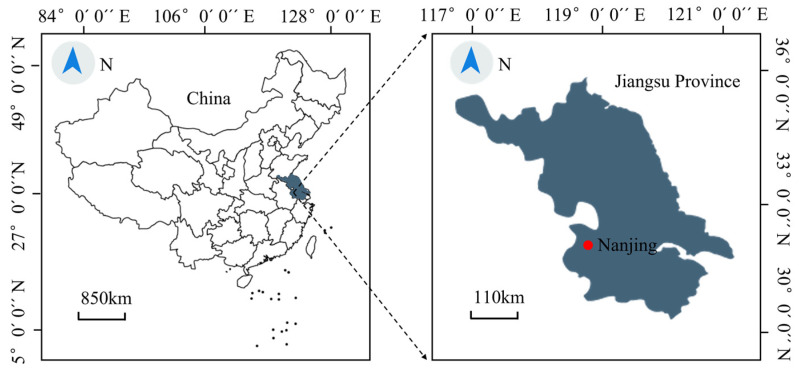
Sampling region.

**Figure 14 sensors-23-03806-f014:**
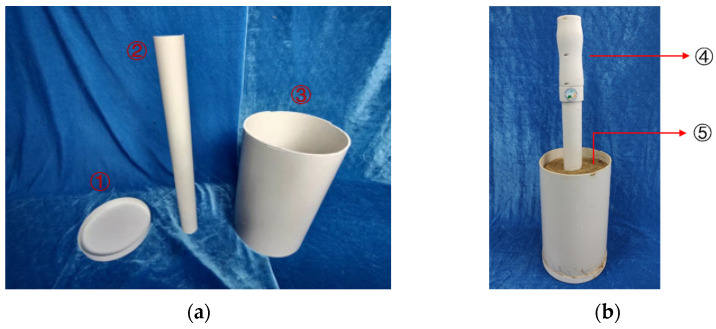
Sensor modification test bucket. (**a**) Modification test barrel parts drawing; (**b**) sensor test chart. ① Lower bottom cover; ② inner tube; ③ outer wall; ④ sensors; ⑤ soil sample to be tested.

**Figure 15 sensors-23-03806-f015:**
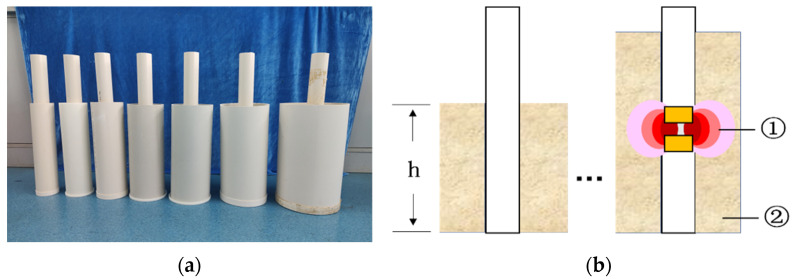
Sensor detection range test. (**a**) Soil samples of different radii to be tested; (**b**) soil samples at different heights to be measured. ① Probe; ② soil.

**Figure 16 sensors-23-03806-f016:**
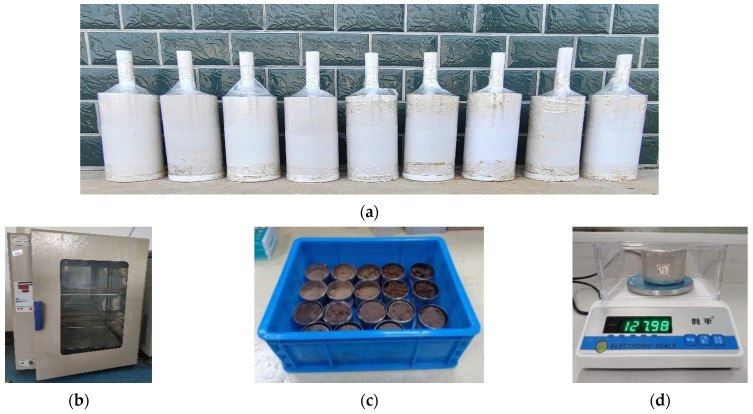
Test plot of moisture measurement model construction. (**a**) Soil samples with different gradient moisture content. (**b**) Hot drum drying box; (**c**) ring knife used to take soil; (**d**) weighing the soil before and after drying.

**Figure 17 sensors-23-03806-f017:**
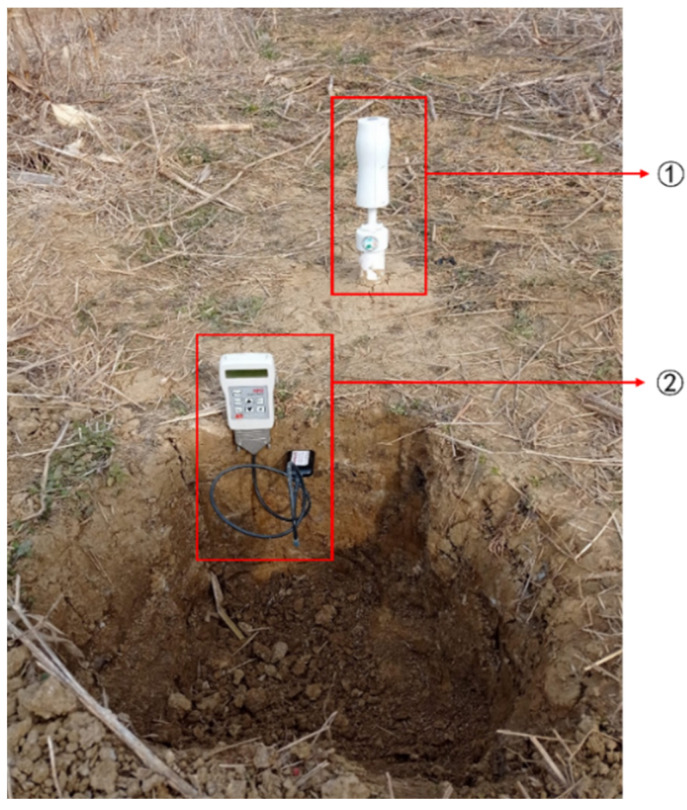
Field performance test chart of self-developed moisture sensor. ① Pull-out soil profile moisture sensor; ② WET-2 commercial moisture sensor.

**Figure 18 sensors-23-03806-f018:**
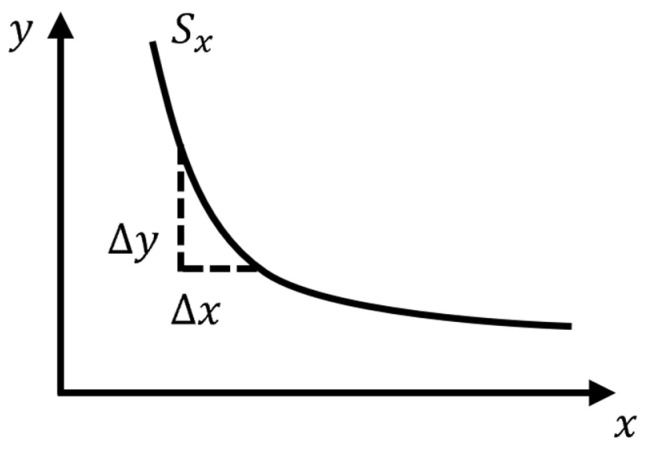
Schematic diagram of the response curve of the output frequency of the sensor probe and the radius or height of the soil sample to be measured.

**Figure 19 sensors-23-03806-f019:**
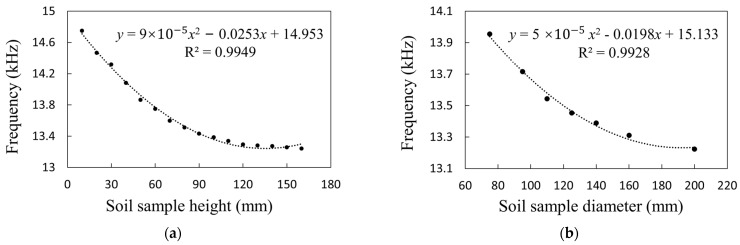
Relationship curve between output signal frequency and soil sample diameter and height. (**a**) Probe output frequency and soil sample diameter; (**b**) probe output frequency and soil sample height.

**Figure 20 sensors-23-03806-f020:**
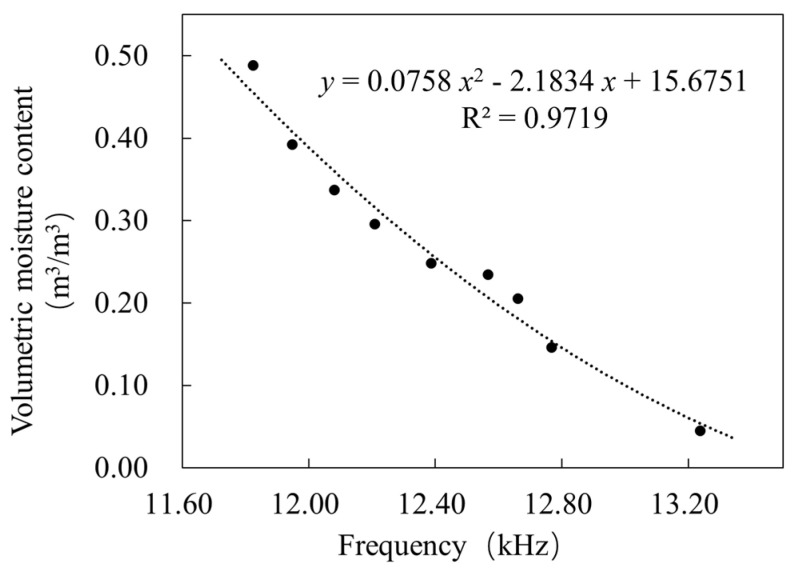
Sensor moisture measurement model.

**Figure 21 sensors-23-03806-f021:**
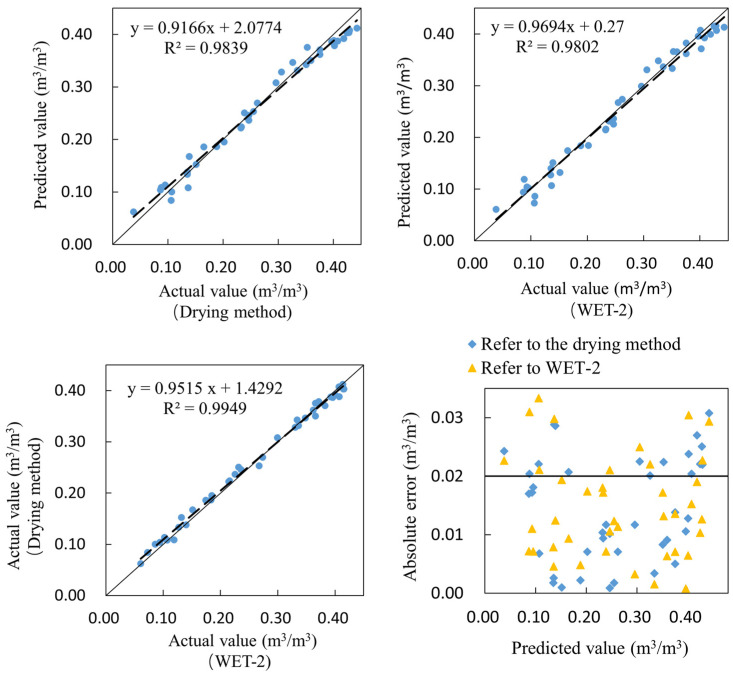
Sensor laboratory accuracy verification.

**Figure 22 sensors-23-03806-f022:**
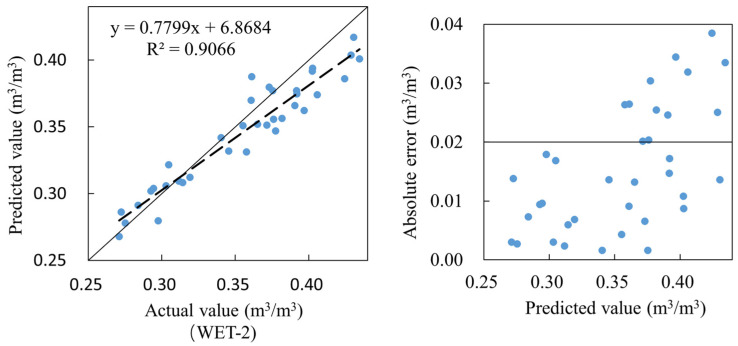
Field accuracy verification of sensors.

**Table 1 sensors-23-03806-t001:** The dielectric constants of the main components of soil.

Composition	Air	Water	Ice	Basalt	Granite	Dry Fertilizer	Dry Sandy	Dry Loam
Dielectric constant	1	78.2	3 (5 °C)	12	7–9	3.5	2.5	2.68

**Table 2 sensors-23-03806-t002:** Indoor measurement accuracy verification test results of self-developed moisture sensor.

Index	Refer to the Oven-Drying Method			Refer to WET-2		
	RMSE	MBE	Maximum Error	RMSE	MBE	Maximum Error
Value (m^3^/m^3^)	0.017	±0.033	±0.031	0.018	±0.050	±0.033

**Table 3 sensors-23-03806-t003:** Field measurement accuracy verification test results of self-developed moisture sensor.

Index	RMSE	MBE	Maximum Error
Value (m^3^/m^3^)	0.020	±0.009	±0.039

## Data Availability

The dataset used in this research is available upon valid request to any of the authors of this research article.

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
