# Peer review of "A Portable Pull-Out Soil Profile Moisture Sensor Based on High-Frequency Capacitance"

_sensors, 2023, doi:10.3390/s23083806_

Round 1

Reviewer 1 Report

The manuscript entitled A portable pull-out soil profile moisture sensor based on high-frequency capacitance. submitted by the group of Authors, represents a research on portable pull-out soil profile moisture sensor based on the principle of high-frequency capacitance. The maximum detection height for the sensor is 130 mm, the maximum detection radius is 96 mm. The field test was performed to verify measurement accuracy of the sensor in field. The results were compared with WET-2 Commercial Moisture Sensor.

The work was well presented. The discussion supported the results. A referent method was used and real samples were measured. The introduction part should be enriched with recent references.

The conclusions should be brief and concise. Authors should shorten the conclusions and give the most important conclusions of the research together with corresponding results comments and possibly the future perspective/usage at the end.

The English should be corrected by a proof reading service.

In general, the manuscript is interesting and applicable.

Reviewer 2 Report

1-Abstract section can be improved. For example  kısmı biraz daha geliştirlebilir. Other sensors compared can be mentioned. For example, when the study was done.

2- There are 2 material and methods section in this manuscript. why? I couldnt understand. This confusion should be corrected.

3-Maybe, Authors can add some literatures for the introduction section. 

4-The authors also can enlarge the discussing part . 

5-The conclusions section is written well. 

Reviewer 3 Report

This paper proposed a low-cost single-probe pull-out soil profile moisture sensor based on high-frequency capacitance. The authors believe that this sensor has the characteristics of low cost and high accuracy compared to some commercial sensors.  The idea of the paper, the experimental scheme, and the design of the sensor are feasible. However, there are still some questions that need to be answered and revised.

1.       Introduction: There is insufficient discussion on existing research on soil moisture sensors, and relevant research in the past five years should be added.

2.       In addition to the low-cost advantages of sensors. State the novelty of your research at the end of the introduction.

3.       The symbol of the formula in the article should give the unit, such as lines 110-116. The same problem also exists later. In addition, each symbol in the formula should be explained, such as Formula 8-15.

4.       I think Figure 13 is not helpful for the study of this article, and I suggest deleting Figure 13.

5.       The font in the image in the article should be consistent with the text, such as Figures 12, 18, etc.

6.       How is the outer diameter of the test barrel set? Why did the experiment take 75, 90,110, etc. as the outer diameter dimensions? An explanation should be given.

7.       The conclusion or discussion section should provide the applicable range and usage conditions of the sensor.

8.       In addition to low cost, what are the advantages and disadvantages of the proposed sensor compared to other sensors.

9.       The conclusion part simply reported the findings. At the end of the conclusion, your suggestions for the promotion, application, and research of sensors should be given.

Reviewer 4 Report

The manuscript has originality/novelty, is well written providing sufficient background for the tool designed. I suggest accepting it in its current form.

Round 2

Reviewer 3 Report

 The corrections are done.  I think the article is ready to publish.